# Identification of Skin Lesions by Using Single-Step Multiframe Detector

**DOI:** 10.3390/jcm10010144

**Published:** 2021-01-04

**Authors:** Yu-Ping Hsiao, Chih-Wei Chiu, Chih-Wei Lu, Hong Thai Nguyen, Yu Sheng Tseng, Shang-Chin Hsieh, Hsiang-Chen Wang

**Affiliations:** 1Department of Dermatology, Chung Shan Medical University Hospital, No.110, Sec. 1, Jianguo N. Rd., South Dist., Taichung City 40201, Taiwan; missyuping@gmail.com; 2Institute of Medicine, School of Medicine, Chung Shan Medical University, No.110, Sec. 1, Jianguo N. Rd., South Dist., Taichung City 40201, Taiwan; 3Department of Mechanical Engineering and Center for Innovative Research on Aging Society (CIRAS), National Chung Cheng University, 168, University Rd., Min Hsiung, Chia Yi 62102, Taiwan; gskyjob01@gmail.com (C.-W.C.); nguyenhongthai194@gmail.com (H.T.N.); cwlu@mdamo.com (Y.S.T.); 4Director of Technology Development, Apollo Medical Optics, Inc. (AMO), 2F., No. 43, Ln. 188, Ruiguang Rd., Neihu Dist., Taipei City 114, Taiwan; lonesome310160@hotmail.com; 5Department of Plastic Surgery, Kaohsiung Armed Forces General Hospital, 2, Zhongzheng 1st.Rd., Lingya District, Kaohsiung City 80284, Taiwan

**Keywords:** mycosis fungoides, single shot multibox detector, psoriasis, atopic dermatitis, optical coherence tomography

## Abstract

An artificial intelligence algorithm to detect mycosis fungoides (MF), psoriasis (PSO), and atopic dermatitis (AD) is demonstrated. Results showed that 10 s was consumed by the single shot multibox detector (SSD) model to analyze 292 test images, among which 273 images were correctly detected. Verification of ground truth samples of this research come from pathological tissue slices and OCT analysis. The SSD diagnosis accuracy rate was 93%. The sensitivity values of the SSD model in diagnosing the skin lesions according to the symptoms of PSO, AD, MF, and normal were 96%, 80%, 94%, and 95%, and the corresponding precision were 96%, 86%, 98%, and 90%. The highest sensitivity rate was found in MF probably because of the spread of cancer cells in the skin and relatively large lesions of MF. Many differences were found in the accuracy between AD and the other diseases. The collected AD images were all in the elbow or arm and other joints, the area with AD was small, and the features were not obvious. Hence, the proposed SSD could be used to identify the four diseases by using skin image detection, but the diagnosis of AD was relatively poor.

## 1. Introduction

Mycosis fungoides (MF) is a rare disease, unlike common skin cancers, such as squamous cell carcinoma, basal cell carcinoma, and melanoma, which are mainly caused by long-term exposure to the sun. By contrast, MF is special, and its cause is still unclear. In the premycotic phase, it appears as a scaly red rash in areas of the body, which is hard to diagnose as mycosis fungoides. Given that the early symptoms of MF are similar to the foci of psoriasis (PSO) and atopic dermatitis (AD), MF is often mistaken for the later skin disorders, delaying the time for treatment. At present, MF cancer is primarily examined by the use of pathological tissue slices [1,2,3,4], lymphatic imaging examination [5,6,7,8], hematology examination [9], and other invasive detection methods. The examination process is not only complex and time-consuming but also leads to great psychological pressure on the patient. The capability of artificial intelligence (AI) in interpreting images can be used to achieve the recognition level similar to those of traditional inspection methods and the detection accuracy of noninvasive diagnosis.

In 2016, Pomponiu et al. used deep CNN to quickly classify skin cancer. The maximum average accuracy, sensitivity, and specificity values were 95.1 (squamous cell carcinoma), 98.9 (light keratosis), and 94.17 (squamous cell carcinoma) [10,11]. In 2017, Esteva et al. classified the clinical images of melanoma or nevus by using machine vision. From the initial clinical examination, through dermoscopy analysis and histopathological examination, they classified skin lesions and skin diseases into three major categories, malignant, benign, and nontumor. Compared with the classification by skin experts, machine vision has reached the equivalent of diagnosis by dermatologists as shown by the classification results [12]. In 2018, Haenssle et al. demonstrated a CNN’s diagnostic performance with a large international group of 58 dermatologists. Irrespective of any physicians’ experience, they may benefit from the assistance by a CNN’s image classification [13]. In 2019, Tschandl et al. successfully applied the convolutional neural network (CNN) to diagnose skin cancer. To verify the effectiveness of this model, the results were compared with the professional interpretation of 95 dermatologists, and the results showed that the CNN could interpret skin images as accurately as dermatologists [14]. In 2019, Hekler et al. explained that the combination of human skills and AI CNN to detect suspicious skin cancer images is better than the independent method. The combined method achieved an accuracy of 82.95%, while the accuracy of AI alone is 81.59%, compared to 42.94% with human judgment alone. These results showed that the combination of human skills and AI achieves better results than the independent results of these two systems [15].

SSD mainly predicts from multiple feature maps. Each feature map is used to detect objects of multiple scales; non-maximum suppression by generating the final test result. In 2019, Faaiza et al. demonstrated a novel method based on SSD and level set method for automated melanoma detection [16]. This research is performed to propose a rapid and universal disease detection method, which can be applied to images of skin diseases. The single-step multiframe detector (SSD) was considered as an example for demonstration. The authors hope to use the interpretation ability of AI in images to achieve the same level of recognition as those of traditional inspection methods and the detection effect similar to noninvasive diagnosis.

## 2. Experimental Section

### 2.1. Sample Preparation

In this study, in cooperation with the Chung Shan Medical University Hospital (CSMUH), the CSMUH Dermatology Department provided 948 skin images as training images, including 215 MF, 421 Pso, 139 AD, and 173 normal skin images. The images were divided into four types according to the doctor’s tissue analysis and grouped into separate training images. The SSD was constructed through a CNN, and 292 test images were prepared to evaluate the accuracy of the diagnosis by the model.

### 2.2. Ethical Statement

The Institutional Review Board of the Chung Shan Medical University Hospital approved the forms for the informed consent and the study protocol (IRB Number CS19168). All participants gave their informed consent. All approaches were performed in accordance with the regulations and relevant guidelines. The experiment design and process were assessed by the ethics committee of the Chung Shan Medical University Hospital and did not involve ethical experiments.

### 2.3. Clinical Features

The skin images used in this study were from by the Department of Dermatology, Chung Shan Medical University Hospital. We used a digital camera (E-M10 Mark III/Olympus) for skin photography. The doctor personally performs image capture on the patient’s lesion location. The disease was classified based on the doctor who observed the pathological tissue section of the patient and confirmed that the patient showed symptom. The patient’s photo and tissue section image are shown in Figure 1.

Figure 1d is a tissue of the patient with MF. No changes in the stratum corneum was observed. The yellow arrow points to the lymphoma cells. Generally, the lymphocytes are in the dermis, while the lymphoma cancer cells in patients with MF will rush upward into the epidermis. Figure 1e is a tissue of a patient with Pso. The stratum corneum (indicated by the red arrow) became very thick because of the incomplete keratosis. The blood vessels of the papillary dermis expanded and became very complex. Figure 1f is a tissue of a patient with AD. The green arrow in Figure 1f shows that the dermal papillary layer was enlarged, and the light green arrow shows where the plasma penetrated into the epidermis. The information of these pathological slices are listed in Table 1. The redness of the skin of the three diseases could be inferred to be unrelated to the increase in the red blood cells, which is the general perception. The stratum corneum and the granular layer showed some differences. We also used the optical coherence tomography (OCT) analysis to check the ground truth samples of this research. The detailed OCT study is shown in the Appendix A.

### 2.4. Single Shot Multibox Detector (SSD)

A total of 948 skin lesion images were collected. The trained skin images included 215 MF, 421 Pso, 139 AD, and 173 normal skin. The detailed image data amplification method is described in the Appendix A. The SSD (SSD-HS, Hitspectra Intelligent Technology Co., Ltd., Taiwan), which is a fast, deep CNN model architecture, was used to build a diagnosis system based on AI [17,18,19]. The category target detector is shown in Figure 2.

With the input of SSD 300 × 300, to generate more layers with smaller scales for facilitating the fusion of multilayer features, Conv7, Conv8_2, Conv9_2, Conv10_2, and Conv11_2 were extracted from the new convolutional layer as the feature maps for detection. In addition to the Conv4_3 layer, a total of six feature maps with sizes of (38,38), (19,19), (10,10), (5,5), (3,3), and (1,1) were extracted. Among them, in the flowchart of the SSD, the first convolutional layer Conv4_3 after the VGG16 model, was used as the first feature map for detection. The scale of the Conv4_3 layer was different from the subsequent convolutional layer, so L2 Normalization was performed. After normalization, the features between different dimensions were compared to improve the accuracy of the classifier and the convergence speed. After the features of five convolutional layers were extracted, non-maximum suppression was used to eliminate the redundant frames to find the best bounding box and generate the final prediction result. The detailed SSD architecture is shown in the Appendix A. After using the SSD to learn the training image set, 300 independent test images were used to evaluate the performance of the trained model. When the model detector detected skin lesions from the input data of the test image, the disease name (PSO, AD, MF, and Normal) and a rectangular frame in the skin image were displayed to surround the lesion area and analyze the evaluation results with the following indicators: Recall, precision, and F1-score. Classifier precision determines how many of the positive categories of all the samples are true positive. The recall rate indicates how many of the true-positive category samples are judged as positive category samples by the classifier [20,21,22]. F1-score is the harmonic mean of the accuracy and recall rate. Macro-average refers to the arithmetic mean of each statistical indicator value of all categories. The micro-average is used to establish a global confusion matrix for each model example in the dataset without category and then calculate the corresponding indicators.

## 3. Results and Discussion

Figure 3 shows the result of using SSD to diagnose skin lesion images. The SSD used the marked bounding orange box to identify MF, blue box to identify Pso, blue-gray box to identify AD, and light-orange box to identify normal skin. The green box indicates the ground truth, which was manually selected.

Comparison of the ground truth and bounding box showed whether SSD could diagnose MF. Table 2 shows the results from diagnosing 292 skin disease images by using SSD. The experimental results showed that 10 s elapsed for the SSD to analyze 292 skin lesion test images and correctly detect and classify 273 images. The images included 139 Pso, 40 AD, 50 MF, and 63 Normal images. The diagnosis result is displayed in a confusion matrix, in which 273 images were correctly diagnosed. Thus, the SSD diagnosis accuracy rate was 90%. A total of 135 PSO, 29 AD, 46 MF, and 60 Normal images were correctly diagnosed.

The accuracy rate of the SSD diagnosis was 93%. The sensitivity values of the SSD to diagnose lesions were 96%, 80%, 94%, and 95% for Pso, AD, MF, and normal skin, with corresponding accuracy rates of 96%, 86%, 98%, and 90% and F1-scores of 96%, 83%, 96%, and 92%. The possible reasons for the low sensitivity of SSD to AD (80%) were as follows: fewer AD images; smaller lesion area; and the location is on the curved side of the arm or elbow. The brightness of the image affected the diagnosis of the SSD, and the calculated analysis indices of the SSD in detecting skin lesions are listed in Table 3.

For the sensitivity, whether AD or normal, the sensitivity values were quite low probably because of these were mostly pictures of arms/elbows at relatively low magnification, so that the entire arm was part of the picture. For the accuracy, the normal image had the highest result, followed by MF. For the F1-score, the best judgment of the SSD was shown in PSO and MF, which may be due to the large amount of image data and obvious symptoms.

In terms of sensitivity and accuracy, MF and PSO were both remarkably higher than those of AD and the normal skin. PSO is more common in terms of its own imaging features, and desquamation and patches were found on the skin. Patients with MF often have large areas of affected skin because of the spread of lymphoma cells. From some photos of the patient, the whole body was symptomatic, resulting in the difficulty in distinguishing which part of the disease spread. Similar to a normal cancer, cancer cells would spread to the entire body and organs. The most sensitive part was MF, which showed that among all the results of true MF, the model predicted a high proportion of MF, which helped assess this disease. F1-score is the result of comprehensive sensitivity and accuracy, which was 96% for MF (second only to 96% of PSO). Using the SSD model to identify skin images could help doctors to quickly diagnose skin lesions. If more skin disease images are available in the future, the database of this study can be more complete. For example, photos of severe and well-cured PSO would be classified in different tags, and the number of images of each category tag will increase, and the skin image recognition model of this study will be more complete and more accurate. The proposed method is expected to assist doctors to use the identification method of the image database to allow patients to obtain the corresponding treatments for different symptoms.

## 4. Conclusions

The constructed SSD model has at least one order of magnitude higher number of a priori boxes than existing methods to predict the location, scale, and aspect ratio of the sample. The system could promote early detection in practice, and better predictions will be achieved in the near future. Therefore, the use of SSD technology in skin lesions can help clinicians quickly diagnose skin lesions. In the future, a larger amount of images can be established through the SSD image diagnosis system developed in this study to help clinicians use the database to more efficiently classify and treat skin lesions.

## Figures and Tables

**Figure 1 jcm-10-00144-f001:**
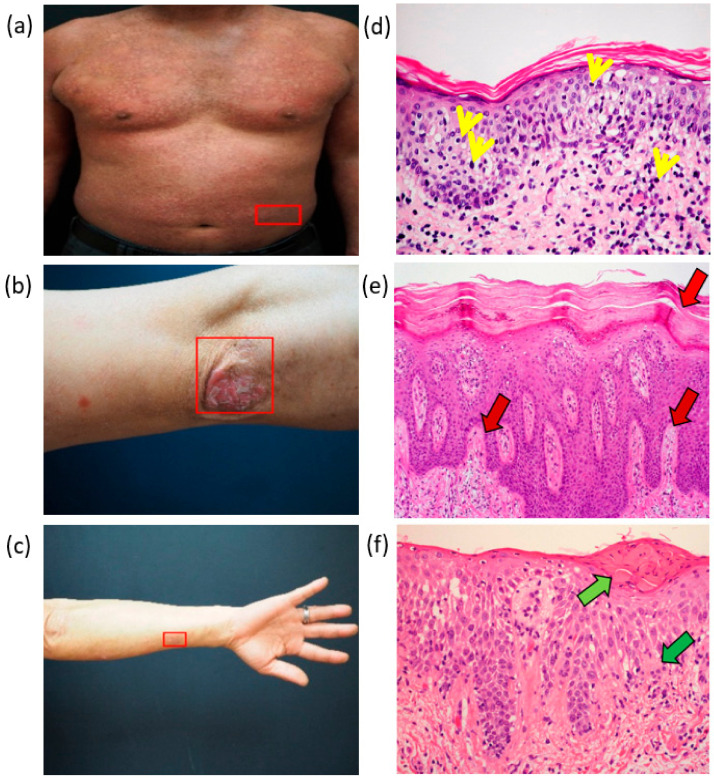
(**a**) Mycosis fungoides (MF), (**b**) psoriasis (PSO), and (**c**) atopic dermatitis (AD) and (**d**–**f**) corresponding pathological tissue slices.

**Figure 2 jcm-10-00144-f002:**
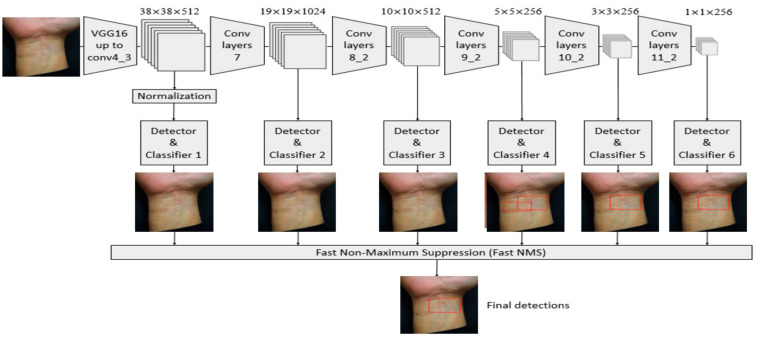
Image flow chart of the single-shot multibox detector (SSD) for MF detection.

**Figure 3 jcm-10-00144-f003:**
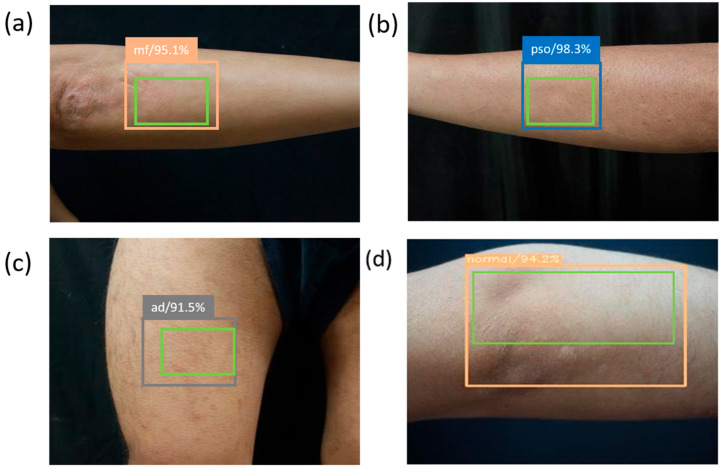
The SSD model predicts the classification of skin disorders. All green boxes are ground truth, and the numbers in the label indicate the probability of being judged as the skin disorder in the box. (**a**) is the MF image, which displays the orange bounding box surrounding the lesion area; (**b**) is the image of the PSO, with the blue bounding box surrounding the lesion area; (**c**) is the skin image of the AD area, with the blue-gray bounding box surrounding the lesion area; and (**d**) is a normal skin image.

**Table 1 jcm-10-00144-t001:** Comparison of the pathological changes in mycosis fungoides (MF), psoriasis (PSO), and atopic dermatitis (AD).

Pathology	MF	PSO	AD
Stratum corneum	No change	Thickened	Thickened
Stratum granulosum	No change	Fewer	Increased
Keratinocyte	Cell edema	Cell edema	Cell edema
Dermis	Lymphocytes gathered at the border of the epidermis, fibrosis	Mastoid dermis, dilated blood vessels	Mastoid dermis, edema, fibrosis
Lymphocytes	As the number increased, abnormal lymphocytes rushed up into the epidermis	Abnormal lymphocytes rushed up into the epidermis	Uncertain

**Table 2 jcm-10-00144-t002:** Diagnosis of 292 skin images by using SSD, in which a confusion matrix is used to display the results. The accuracy of SSD diagnosis is 93%, and the sensitivity of SSD diagnosis of skin lesions is arranged in the order of PSO, AD, MF, and Normal and the corresponding percentages were 96%, 80%, 94%, and 95%. The corresponding accuracy rates were 96%, 86%, 94%, and 95%.

Skin Disease	Results of Prediction by SSD			
Images	PSO	AD	MF	Normal
PSO	134	3	0	2
AD	3	32	1	4
MF	1	1	47	1
Normal	2	1	0	60

**Table 3 jcm-10-00144-t003:** Diagnosis of skin image by the SSD based on the diagram of the calculated analysis indices.

	Sensitivity (%)	Precision (%)	F1-Score (%)	Accuracy (%)
PSO	96	96	96	
AD	80	86	83	
MF	94	98	96	
Normal	95	90	92	
Total				93

## Data Availability

Restrictions apply to the availability of these data. Data was obtained from Chung Shan Medical University Hospital and are available from Dr. Yu- Ping Hsiao with the permission of Chung Shan Medical University Hospital.

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
