# Peer review of "Identification of Skin Lesions by Using Single-Step Multiframe Detector"

_jcm, 2021, doi:10.3390/jcm10010144_

Round 1

Reviewer 1 Report

Hsiao et al. present a manuscript on automated skin lesion detection and classification using a single-step multiframe detector. While high-level skin lesion classification using convolutional neural network has been shown in several studies, the use of an SSD in this context presents a novel stategy.

However, I think the mansuscript will have to be modified quite substantially to achieve publication in the envisaged special issue.

Comments and questions

Title/Abstract:

1) The title does not reflect the content of the study very well.

2) Also, the abstract contains information that does not reoccur in the main text, but lacks important basic information about the methods that were employed and the setup of the study.

Introduction:

1) The references need to be rechecked carefully, and their contents have to be described more carefully. For instance, the quoted reference by Brinker et al. is a review article. The Esteva et al. study lesions were also classified as melanoma or nevus. On emight also quote publications from the Haenssle group. References to the single-shot detector are missing altogether.

2) The SSD method must be introduced in the introduction.

3) The language issues in this section should be resolved.

4) It is unclear why mycosis fungoides, which should be defined as a cutaneous lymphoma, was chosen as the "main" comparator, especially since the special issue is dealing with chronic inflammatory skin disorders such  as psoriasis and atopic dermatitis.

Experimental Section

1) Information on how the images were taken and on which images were chosen is missing.

2) The information provided in Figure 1 and the text referring to this Figure is not strictly required.

3) The purpose of the section on OCT is not clear to me. OCT is an interesting technique, but in what way does this section contribute to the "story" of the manuscript?

4) The information on how the SSD and the bounding boxes are generated is quite detailed for an audience of non-informaticians. Since much of this information is already provided in the supplementary material, this section could be shortened here. In line 196, the second "a a priori box" should probably be replaced by "ground truth"?

5) The metrics that were used to describe the results should be introduced here.

Results and Discussion:

1) Major point: The authors do not provide details on the composition of the datasets. In the Introduction, they state that the chosen lesions may resemble each other very closely at early stages, but the discussion seems to indicate that many of the lesions that were chosen for analysis were fairly advanced. More information would be required to be able to judge the accuracy that was achieved: Would the diagnosis have been difficult for the dermatologist in many or at least some of the cases, or were they mostly unequivocal?

2) Related point: At the present stage, the study indicates that an SSD may by used to diagnose various skin lesions. However, as the authors state, especially for MF, quite advanced lesions were chosen for analysis. Thus, this study does not reflect a situation where the algorithm could be used to assist dermatologists in the early detection/differential diagnosis of these lesions. This should be stated clearly in the discussion.

3) Some aspects need to be clarified further. For instance, in lines 243/244, the authors state that sensitivity values of AD and normal tissue, the values were quire low probaly because of the shadowed part. Here you should explain that these were mostly pictures of arms/elbows at relatively low magnification, so that the entire arm was part of the picture.

Conclusions:

1) The conclusion is rather long and resembles a discussion section. In my opinion, this section should be shortened to the tow or three sentences containing the key points.

2) The information that the SSD model took 10 seconds to classify the 300 test images should also be included in the results section - have I overlooked it there?

3) The authors should take care not to overinterpret their results - as mentioned previously, the study may provide a proof of principle, but in its current state comparisons to other methods and statements with respect to the helpfulness of such systems in the clinic seem a bit premature.

Supplementary data:

1) Large parts of the supplementary data are redundant with information provided in the main manuscript already.

Reviewer 2 Report

I have got big objections about the substantial things: mycosis fungoides is not cancer. Putting it together with the BCC, SCC (in the introduction) is a big abuse. The basis for the diagnosis of MF is histopathological examination still, which must be repeated several times...

The use of some clinical descriptions like "lumps" are not dermatological terms.

References needs to be revised. 

Reviewer 3 Report

This manuscript encompasses a detailed description of the training of a  CNN for the diagnosis of MF, Psoriasis, and Atopic Dermatitis. I think that the article needs significant improvements before resubmitting  to this, or another journal.

Primarily, I would strongly suggest to the authors to have a native English speaker go over the article before resubmitting. 

Second, I sincerely did not understand the addition of the OCT part. It does not add anything to the article and the CNN part.

Third, Both the training and test dataset were extremely small in numbers.

The results need a more cohesive presentation, while the methods are extremely detailed and verbal in the training of a CNN, a well established method in the literature.

The authors did not compare their CNN with clinicians, while they also mix sensitivity and specificity with 'accuracy rate'. An ROC curve would be suggested in order to display what they wish.

Round 2

Reviewer 1 Report

I really appreciate the fact that the authors went to a lot of trouble in order to improve their manuscript by re-visiting all their data.

I only have a few minor suggestions:

1) The title has improved, but you might consider changing it so that it encompasses all the lesions that you analyzed, not only mycosis fungoides (e.g. identification/classification of skin lesions...)

2) Maybe you could transfer the explanation for the added value of the OCT to the text.

3) The conclusion is still a bit long - readers might appreciate it if you could shorten it further to clarify what the main message is in your opinion.

Author Response

Reviewer 1:

I really appreciate the fact that the authors went to a lot of trouble in order to improve their manuscript by re-visiting all their data.

I only have a few minor suggestions:

1) The title has improved, but you might consider changing it so that it encompasses all the lesions that you analyzed, not only mycosis fungoides (e.g. identification/classification of skin lesions...)

Reply:

We appreciate your comments. We corrected the title as “Identification of skin lesions by using single-step multiframe detector” in this manuscript.

2) Maybe you could transfer the explanation for the added value of the OCT to the text.

Reply:

We move this “OCT section” into “Supplementary Information”. The second reviewer suggest to move this part.

3) The conclusion is still a bit long - readers might appreciate it if you could shorten it further to clarify what the main message is in your opinion.

Reply:

We shorten the content of "conclusion".

Reviewer 3 Report

Dear authors, congratulations on the major improvements on your article. However, I still think that the OCT part obscures your message. I would remove it completely. 

Author Response

Reviewer 3:

Dear authors, congratulations on the major improvements on your article. However, I still think that the OCT part obscures your message. I would remove it completely.

Reply:

We appreciate your comment. We move this “OCT section” into “Supplementary Information”.